# Differences in the Response to DNA Double-Strand Breaks between Rod Photoreceptors of Rodents, Pigs, and Humans

**DOI:** 10.3390/cells9040947

**Published:** 2020-04-12

**Authors:** Florian Frohns, Antonia Frohns, Johanna Kramer, Katharina Meurer, Carla Rohrer-Bley, Irina Solovei, David Hicks, Paul G. Layer, Markus Löbrich

**Affiliations:** 1Radiation Biology and DNA Repair, Technical University of Darmstadt, 64287 Darmstadt, Germany; Meurer.Katharina@gmail.com (K.M.); lobrich@bio.tu-darmstadt.de (M.L.); 2Plant Membrane Biophysics, Technical University of Darmstadt, 64287 Darmstadt, Germany; Antonia.Frohns@gmail.com; 3Developmental Biology and Neurogenetics, Technical University of Darmstadt, 64287 Darmstadt, Germany; JohannaksKramer@googlemail.com (J.K.); Layer@bio.tu-darmstadt.de (P.G.L.); 4Division of Radiation Oncology, Vetsuisse Faculty, University of Zurich, CH-8057 Zürich, Switzerland; crohrer@vetclinics.uzh.ch; 5Biozentrum, Biology-II, University of Munich (LMU), 82152 Planegg-Martinsried, Germany; Irina.solovei@lrz.uni-muenchen.de; 6Département de Neurobiologie des Rythmes, Institut des Neurosciences Cellulaires et Intégratives, Université de Strasbourg, 67084 Strasbourg, France; photoreceptor67@hotmail.com

**Keywords:** genome editing, CRISPR-Cas9, DNA double-stranded break, DNA repair, ATM, KAP1, chromatin, retina, photoreceptors, inherited retinal dystrophies

## Abstract

Genome editing (GE) represents a powerful approach to fight inherited blinding diseases in which the underlying mutations cause the degeneration of the light sensing photoreceptor cells of the retina. Successful GE requires the efficient repair of DNA double-stranded breaks (DSBs) generated during the treatment. Rod photoreceptors of adult mice have a highly specialized chromatin organization, do not efficiently express a variety of DSB response genes and repair DSBs very inefficiently. The DSB repair efficiency in rods of other species including humans is unknown. Here, we used ionizing radiation to analyze the DSB response in rods of various nocturnal and diurnal species, including genetically modified mice, pigs, and humans. We show that the inefficient repair of DSBs in adult mouse rods does not result from their specialized chromatin organization. Instead, the DSB repair efficiency in rods correlates with the level of Kruppel-associated protein-1 (KAP1) expression and its ataxia-telangiectasia mutated (ATM)-dependent phosphorylation. Strikingly, we detected robust KAP1 expression and phosphorylation only in human rods but not in rods of other diurnal species including pigs. Hence, our study provides important information about the uniqueness of the DSB response in human rods which needs to be considered when choosing model systems for the development of GE strategies.

## 1. Introduction

Retinal dystrophies which are caused by inherited genome mutations can result in the degeneration of rod and cone photoreceptors (PRs) as the light-sensing cells of the retina. In case of rod-cone dystrophies, the underlying mutation causes the initial dying of rods which is followed by the secondary degeneration of cones and thus results in complete blindness [1]. Hence, rod-specific genetic mutations represent a promising target for gene correction via genome editing (GE) tools like CRISPR-Cas9 in order to prevent inherited blindness.

The CRISPR-Cas9 technique relies on the endonuclease Cas9 to generate double-stranded breaks (DSBs) at a specific site of the genome with the subsequent repair of the induced breaks by the endogenous DSB repair machinery. Additionally, Cas9 lacking endonuclease activity can be used to selectively perturb genome expression by blocking transcription [2,3]. Repair of Cas9-induced DSBs takes place either by non-homologous end-joining (NHEJ) which is available in all cell cycle phases or by homologous recombination (HR). Unlike NHEJ, HR uses a sister chromatid as a template and thus repairs DSBs in post-replicative regions of DNA in S and G2 phase [4]. However, in order to make use of the higher accuracy of HR over NHEJ, there are efforts to establish high fidelity GE by HR even in postmitotic cells. In fact, this can be achieved by the combined delivery of a donor template together with the CRISPR-associated endonuclease Cas9 [5]. Using this approach, Bakondi et al. could successfully modify mutated alleles in postmitotic rods of newborn pups of a rat model for rod degeneration-based blinding diseases [6]. However, up to now, this approach is only successful in rods of young (postnatal) animals, hindering its translation to human patients, as retinal dystrophies are only diagnosed at late stages of human development. However, various research groups are working on ways to further improve the HR efficiency in postmitotic cells, e.g., by using molecules that are inhibiting NHEJ [7].

In order to improve the HR efficiency in postmitotic cells, detailed knowledge about the particularities of individual cell types with regard to their DNA damage response and their DSB repair capacities is of great importance. We and others have previously shown that fully differentiated rod PRs of adult mice repair DSBs very inefficiently [8,9]. Of note, these cells fail to efficiently express the DNA damage response kinase ataxia-telangiectasia mutated (ATM), although an alternative splicing form seems to exist [9]. In addition, the DSB-induced ATM autophosphorylation is strongly impaired in adult mouse PRs, suggesting a strong diminution of ATM kinase activity [8]. Additionally, fully differentiated rods show reduced expression levels of the ATM target protein Kruppel-associated protein-1 (KAP1) and a strong defect in the DSB-induced and ATM-dependent activation of KAP1 [8]. KAP1 has been described in cultured cells where it localizes to heterochromatin and represents a barrier to DSB repair that is alleviated after its ATM-dependent phosphorylation at serine 824 (pKAP1) [10,11,12,13]. However, in retinal cells, KAP1 localizes to euchromatin [8] and it is unknown how this affects DSB repair. Some studies have shown that KAP1 is recruited to sites of DSBs, suggesting that it is not only a barrier to DSB repair but might exert an active function during this process [14,15]. Consistent with this idea, we have observed KAP1 recruitment to laser track-induced DSBs in undifferentiated rods of 4 day old mice (postnatal day 4, P4) which, in contrast to adult mice, show robust KAP1 expression and efficient DSB repair [8]. Hence, it is possible that the reduced level of KAP1 expression and the strong diminution of its ATM-induced activation are causative for the DSB repair defect observed in rods of adult mice.

The DSB repair defect that arises during rod differentiation correlates with another process, called nuclear inversion, that takes place in rod nuclei of nocturnal mammals during their differentiation. Whereas in conventional nuclei euchromatin occupies the nuclear interior and heterochromatin is adjacent to the nuclear periphery, these two chromatin classes have inverted arrangements in mouse rods [16,17]: euchromatin is adjacent to the nuclear periphery and heterochromatin is concentrated in the nuclear interior where it forms a globule with a high chromatin density [18]. Nuclear inversion is a process driven by release and coalescence of heterochromatin from the nuclear lamina during rod differentiation [18,19] as a result of deficiency of two tethers of the peripheral heterochromatin, lamin B receptor (LBR)-dependent, and lamin A/C-dependent tethers [20]. Importantly, the striking density of the rod internal heterochromatin globules converts them into micro-lenses, which reduce the scattering of light propagating through the retinae of nocturnal animals, thereby improving their vision [17,18,21]. In contrast, rods in diurnal mammals have conventional nuclear architecture, maintained by expression of either LBR or LAC tethering heterochromatin to the lamina [20]. Although inverted rod nuclei remain overtly functional [17], they exhibit an altered epigenetic landscape [22] and a changed expression of some common proteins [23]. This might explain the downregulation of ATM and KAP1, which in turn causes defects in DNA damage signaling and DSB repair in adult rods [8,9]. This is supported by the fact that undifferentiated rod progenitors (e.g., at P4), which still possess conventional nuclei, have a normal DSB repair efficiency [8]. If this interpretation is correct, conventional rod nuclei of diurnal mammals would not exhibit a DSB repair defect. This would question the suitability of mice as animal models for the development of GE strategies in humans.

In the present study, we investigated the DNA damage response and the efficiency of DSB repair in rods of various species to assess their suitability as model systems for GE strategies in humans. We analyzed KAP1 expression levels and ATM-dependent KAP1 phosphorylation after DSB induction by ionizing radiation (IR), the DSB repair capacity, and the accumulation of the p53-binding protein 1 (53BP1) at DSBs as an additional read-out for DSB-induced signaling events [24,25]. The analysis was carried out in various rodent species including genetically engineered mice whose rods show a non-inverted chromatin organization by ectopic Lbr expression [20], in pigs, monkeys, and humans. Our results show that the DSB repair capacity of rods correlates with the level of KAP1 expression and its ATM-dependent phosphorylation. Strikingly, only rods from primates but not rods from pigs or rodents exhibit robust KAP1 expression and phosphorylation, a finding important for the design of GE strategies to prevent rod degeneration-based blinding diseases.

## 2. Materials and Methods

### 2.1. Animal Irradiation (In Vivo) and Tissue Isolation

Adult C57BL/6NCrl wild-type (WT) mice were obtained from Charles River Laboratories. Lbr-TER mice were obtained from Irina Solovei at the LMU Munich [16]. Arvicanthis ansorgei were obtained from David Hicks at the Université de Strasbourg [26]. X-irradiation of these animals was carried out using a X-Rad 320 (Precision X-ray, Inc., North Branford, CT, USA). Radiation settings were 250 kV, 10 mA, and a dose rate of 1.11 Gy/min. Physical dosimetry with tissue equivalent material confirmed that the variation in dose across the mice was less than 10%.

Pigs were obtained from local breeders in Zurich (Switzerland). Irradiation was carried out at the University of Zurich. Irradiation for all pigs was planned on a non-contrast CT-study, using one pig as representative for the group. In brief, the pig was immobilized in sternal recumbency with in an individually shaped vacuum cushion (BlueBag BodyFix, Elekta AB, Stockholm, Sweden) under general anesthesia. A 1 cm soft-tissue equivalent bolus was placed on the animals face to ensure dose homogeneity at the surface. On the CT-dataset, both eyeballs were contoured with a 3D-contouring tool and the according diameter of the eyes. An additional safety margin of 7 mm was added to allow for set-up error, resulting in a planning target volume (PTV). Computer-based treatment planning was performed with Eclipse External Beam Planning system version 10.0 (Varian Oncology Systems, Palo Alto, CA, USA), applying heterogeneity correction, and AAA-algorithm (10.0.28). To experimentally induce a standardized amount of DNA damage, both eyes were irradiated concurrently, using parallel-opposed fields. Treatment was delivered with a 6 megavolt (MV) linear accelerator (Clinac iX, Varian, Palo Alto, CA, USA) with high accuracy and precision in target localization. Before irradiation, image-guidance (IGRT) using kilovolt (kV)-kV orthogonal radiographs was used for treatment verification. The dose of 1 Gy was prescribed to the ICRU reference point, which was defined as a representative point in the planning target volume on the 100% isodose line.

At the end of the repair time, animals were sacrificed, and their eyes were removed and placed in 4% (*v*/*v*) neutral-buffered formalin for 16 h. For pigs, eyes were opened using a scalpel and 9 mm biopsy punches were taken from the retinae and the surrounding tissue. Biopsies were placed in 4% (*v*/*v*) neutral-buffered formalin for 16 h. Fixed tissues were embedded in paraffin and sectioned at a thickness of 4 μm for immunostaining.

A 72-year-old patient received a bulbectomy at the Universitätsklinikum Giessen. The eye was directly placed in Neurobasal medium (Gibco/Life Technologies, Darmstadt, Hessen, Germany) before its opening with a scalpel. Retinal explants were taken using 9 mm biopsy punches and directly placed in 6-well plates containing 2 mL of Neurobasal medium (Gibco) supplemented with 5% B27 plus supplement (Gibco), 1% l-glutamine (Seromed, Berlin, Germany), 1% Non essential amino acids (Gibco) and 1% Pen/Strep (Gibco). Retinal explants were then directly taken to the TU Darmstadt were irradiation was carried out using a X-Rad 320 (Precision X-ray, Inc., North Branford, CT, USA). Radiation settings were 250 kV, 10 mA, and a dose rate of 1.11 Gy/min. At the end of the repair time the explants were placed in 4% (*v*/*v*) neutral-buffered formalin for 16 h. Fixed tissues were embedded in paraffin and sectioned at a thickness of 4 μm for immunostaining.

All animal experiments were approved by the regional board of Darmstadt (WT, Lbr-TER, and Arvicanthis), the regional board of Zurich (pigs) or Giessen (human retina). Retinae of macaque (*Macaca fascicularis*), marmoset (*Callithrix jacchus*), and sudanian grass rat (Arvicanthis; used for the stainings presented in Appendix A) were post mortem material kindly donated by Dr. Leo Peichl, MPI for Brain Research (Frankfurt, Germany).

### 2.2. Immunofluorescence Analysis of Tissues

For paraffin embedded tissues, protocols were as follows: After dewaxing in xylene and rehydration, sections were incubated in citrate buffer for 1 h at 95 °C for antigen retrieval. Sections were encircled with a liquid blocker (PAP PEN; Kisker Biotech, Steinfurt, Germany), incubated with primary antibodies for 5 h at 37 °C, washed three times with PBS-T (0.1% Tween 20 in PBS) for 10 min each, and incubated with secondary antibody for 2.5 h at room temperature in the dark. After three washes with PBS-T for 5 min each, 4′-6-Diamidin-phenylindol (DAPI) staining was performed for 10 min (0.2 μg/mL DAPI in PBS; Sigma Aldrich, Taufkirchen, Germany). After a final wash in PBS, sections were mounted in mounting medium (H-1000; Vector Laboratories, Burlingame, CA, USA) and sealed with nail polish. For immunostaining of retinae from sudanian grass rat after fixation with 4% formaldehyde and cryosectioning, slides with cryosections were air-dried at room temperature for 30 min, rehydrated in 10 mM sodium citrate buffer for 5 min. Antigen retrieval was performed by heating up to 80 °C in 10 mM sodium citrate buffer for 5 min. After brief rinsing in PBS, the slides were incubated with 0.5% Triton X-100 in PBS for 1 h. Both primary and secondary antibodies were diluted in blocking solution (PBS with 0.1% Triton X-100, 1% bovine serum albumin, and 0.1% Saponin); slides with applied antibodies were incubated in dark humid chambers for 12–24 h at RT. Washings between and after antibody incubations were performed with 0.01% Triton X-100 in PBS at 37 °C, 3 × 30 min. For nuclear counterstain, DAPI was added to the secondary antibody solution (final concentration 2 mg/mL). Sections were mounted in Vectashield antifade medium (Vector Laboratories) under coverslips sealed with a colorless nail polish. For immunostaining of monkey retinae, slides with cryosections were air-dried at room temperature for 30 min. After antigen retrieval (see above), sections were stained in the same manner as paraffinated sections (see above).

### 2.3. Antibodies

Antibodies used for immunofluorescence were: gammaH2AX (Millipore, Darmstadt, Germany, 05-636) 1:400, 53BP1 (Santa Cruz, Dallas, TX, USA, sc-22760) 1:500, KAP1 (Abcam, Cambridge, UK, ab22553) 1:400, pKAP1 (Abcam ab133440) 1:200, H3K9me3 (Abcam ab8898) 1:500, H4K8ac (kind gift from Dr. H. Kimura, Osaka University), CAR (Millipore ab15282), 1:500, lamin B (Santa Cruz sc-6217) 1:400, lamin A/C (Millipore #05-714), LBR (Biozol, Eching, Germany, bs-5081R) 1:200, 1:500, Alexa Fluor488 (Invitrogen A11001, A11008) 1:400, Alexa Fluor594 (Invitrogen A11005, A110012) 1:400, Alexa Fluor647 (Dianova, Hamburg, Germany, 715-605-150) 1:400, Dylight 550 (Dianova, A-24421-05).

### 2.4. Image Analysis

Images of the retinae were taken on a confocal microscope (Leica TCS SP5 II) with LAS AF Lite software (Leica, Wetzlar, Germany). Foci counting in the retinae was performed on captured images by eye. All images are maximum intensity projections of image stacks (z = 5–20) with a focus plane distance of 300 nm. The images were arranged using ImageJ software. For pKAP1 signal quantification in P10, P16, P24, and adult WT mice, regions of interest (ROI) were drawn into the outer nuclear layer (ONL), inner nuclear layer (INL), and into the nuclei free space between the INL and ganglion cell layer (GCL) for background correction. Fluorescence signal intensity was measured in the ROIs and the ratio between ONL and INL was calculated after subtracting the background signal. For KAP1 and pKAP1 signal quantification in rods of Lbr-TER mice, Arvicanthis, pigs, and humans, fluorescence signals were measured in single nuclei and in nuclei free spaces of the ONL for background correction. Fluorescence signal intensity was measured in the determined ROIs and the ratio between rods and cones was calculated after subtracting the background signal.

### 2.5. Statistical Analyses

Statistics were performed using GraphPad Prism statistical software (version 6.07). If not indicated otherwise, at least three independent experiments were carried out. Unpaired two-tailed student’s *t*-test was used for testing of γH2AX, 53BP1, KAP1, and pKAP1 quantifications after in vivo irradiation. For all analyses a 95% confidence interval with *: *p* < 0.05, **: *p* < 0.01, and ***: *p* < 0.001 was defined. The number of each experiment is indicated in the figure legends.

## 3. Results

### 3.1. The Defect in KAP1 Phosphorylation Precedes Nuclear Inversion during Rod Development

We previously described a substantial DSB repair defect in rod PRs of adult mice which was not observed in undifferentiated rods at an early postnatal stage (P4). The DSB repair defect correlated with both the nuclear inversion in adult rods (which has not yet started at P4), the downregulation of KAP1 [9] and with the defect in its ATM-dependent phosphorylation [8]. While the time during development at which the nuclear inversion in rods occurs has already been investigated in previous studies [20], the onset time of the emerging defects in KAP1 signaling and DSB repair remained unknown.

Here, we exposed P10, P16, and P24 mice to 1 Gy of X-rays to induce DSBs and analyzed the phosphorylation of KAP1 at ser824 as a typical read-out for active ATM signaling. We observed robust KAP1 phosphorylation at 15 min after IR (but not in unirradiated controls) in the PRs positioned within the outer nuclear layer (ONL) in P10 mice that was strongly diminished in P16, P24 and adult mice (Figure 1A and Appendix A). Nuclear counterstaining revealed progressing nuclear inversions in rod nuclei from P10 to P24 as evidenced by the decreasing numbers of merging chromocenters, brightly stained with DAPI (Figure 1B). Thus, the reduction of KAP1 phosphorylation at P16 precedes the completion of inversion, which in mice requires 6 postnatal weeks [15]. In contrast, cells in the inner nuclear layer (INL) and the ganglion cell layer (GCL) showed strong KAP1 phosphorylation signals throughout all developmental stages (Figure 1A).

To assess the DSB repair capacity in rod PRs, we enumerated the level of γH2AX foci at the three developmental stages (Figure 1B) [8,27,28]. We observed similar foci numbers at 15 min after IR but an increase of residual foci at 24 h from 1 in P10 to 3 in P16 and 5 in P24 mice (Figure 1C). The latter value is comparable to the number of unrepaired DSBs in adult animals [8]. In summary, the loss of KAP1 phosphorylation at P16 correlates with the first emergence of the DSB repair defect, although the maximal repair defect is reached at later developmental stages.

### 3.2. The Defect in KAP1 Phosphorylation Arises Independently of the Loss of LBR Expression

The results above show that the defect in KAP1 phosphorylation in rods at P16 coincides with the downregulation of LBR expression (lam A/C is not expressed in mouse rods) and the subsequent nuclear inversion [20]. To investigate whether the inverted chromatin arrangement is responsible for the defect in KAP1 expression and phosphorylation, we investigated transgenic Lbr-TER mice, in which LBR is ectopically overexpressed [20] in some but not all cells (Lbr transgenically expressed in rods, Lbr-TER rods; see boxes in Figure 1D). Cells expressing LBR maintain a conventional nuclear architecture, whereas non-expressing cells display the WT inverted phenotype [20]. In contrast to cone PRs (which also show conventional chromatin organization; see circles in Figure 1D), Lbr-TER rods are distributed over the entire ONL and not restricted to its apical part. However both Lbr-TER and WT rods exhibited similar levels of KAP1 expression, strongly reduced in comparison to cone PRs (Figure 1D,F). The radiation-induced phosphorylation of KAP1 at ser824 at 15 min after 1 Gy was observed in cones (circles in Figure 1E) but was significantly reduced in both Lbr-TER and WT rods (boxes in Figure 1E,F). Thus, the defect in KAP1 expression and phosphorylation arises independently of the LBR downregulation and nuclear inversion in rods.

### 3.3. The DSB Repair Defect Occurs Independently of LBR Expression but Correlates with the Lack of KAP1 Phosphorylation

The finding that Lbr-TER rods with a conventional nuclear organization have a defect in KAP1 phosphorylation allowed us to investigate the underlying cause of the DSB repair defect in rods. We measured the DSB repair capacity by γH2AX foci enumeration at 24 h after 1 Gy and observed that the ectopic LBR expression causes a slight improvement of the DSB repair capacity compared to WT rods (Figure 1G). However, compared to INL cells, Lbr-TER rods still showed substantially higher numbers of residual DSBs. Thus, our results reveal that the DSB repair efficiency is impaired in mouse rods with a conventional nuclear organization similarly to rods with inverted nuclei [8]. This suggests that the DSB repair defect in mouse rods is independent of chromatin organization but correlates with the lack of KAP1 expression and phosphorylation. We also analyzed the accumulation of 53BP1 at DSBs which was previously shown to be drastically reduced in rods compared to other retinal cells [8,9,24,25]. We observed that both WT- and Lbr-TER rods are substantially impaired in 53BP1 foci formation (Appendix A).

### 3.4. Nuclei of Cones of Diurnal Rodent Arvicanthis Exhibit Inverted Chromatin Arrangement but Robust KAP1 Phosphorylation and Efficient DSB Repair

It was recently shown that, in difference to nocturnal species, diurnal rodents of the genera Arvicanthis contain equal numbers of cones and rods in their retina (Appendix A) [26]. Interestingly, both rods and cones in the sudanian grass rat, Arvicanthis ansorgei, lack expression of LBR and lam A/C and thus possess an inverted arrangement of chromatin (Appendix A). Therefore, studying cones in this species might further help to disentangle possible connections between DSB repair and chromatin arrangement. We compared the levels of KAP1 expression and IR-induced KAP1 phosphorylation within the PRs of Arvicanthis and mice. Similar to mice, rods of Arvicanthis showed a lower staining intensity for KAP1 than the apically located cones and even further diminished IR-induced pKAP1 signals (Figure 2A–C). Having identified a cellular system with robust KAP1 phosphorylation in an inverted chromatin background, we next studied the DSB repair capacity in this cell type. The retinae of irradiated Arvicanthis showed a similar formation of γH2AX foci at 15 min after IR in rods, cones, and cells of the INL (Figure 2D,E). Strikingly, cones were nearly devoid of γH2AX foci at 8 h after IR, whereas rods still showed a robust number of residual DSBs even at 24 h after the treatment. DSB repair in cones was comparable to cells of the INL, showing that efficient DSB repair is possible in cells with nuclear inversion but requires KAP1 phosphorylation. Thus, the data derived from the cones of Arvicanthis strengthen our conclusion that the DSB repair efficiency requires robust KAP1 phosphorylation, while the nuclear inversion has little or no impact.

### 3.5. Non-Inverted Rod PRs of Pigs Show Intermediate Levels of KAP1 Expression and DSB Repair

We next analyzed KAP1 expression and DSB repair in rod PRs of pigs. Pigs are a diurnal species whose rods show un-inverted chromatin due to the constitutive expression of lam A/C [19] and have become an important animal model for research of human inherited blinding diseases [29]. We observed robust KAP1 signals in the ONL which were lower in rods compared to cones (latter ones marked by arrowheads in Figure 3A), although the difference was less pronounced than in mice and Arvicanthis (Figure 3B). Of note, we failed to detect any pKAP1 signal in the pig retinae at 15 min after IR in three different animals, probably due to the lack of antibody specificity.

Next, we analyzed DSB repair by the enumeration of γH2AX foci and observed a similar induction level at 15 min after 1 Gy in rods, cones, and cells of the INL (Figure 3C,D). Compared with mice and Arvicanthis, DSB repair in pigs proceeded more slowly and all cells exhibited a substatial number of unrepaired DSBs at 8 and 24 h post IR. At 24 h, and less pronounced already at 8 h, rods showed higher levels of residual DSBs compared to cones and cells of the INL. Thus, a rod-specific DSB repair defect can also be observed in pigs but is less pronounced than in rodents. The reduction in the rod-specific DSB repair defect from mice to pigs is consistent with the observation that the differences in KAP1 expression levels between rods and cones are smaller in pigs than in rodents.

To further compare the DSB response in pig rods with the situation in mice, we assessed the formation of 53BP1 foci at IR-induced DSBs. At 15 min after IR, all cells of the INL showed robust 53BP1 focus formation while rods were almost completely devoid of 53BP1 foci (Appendix A). Cones also displayed 53BP1 foci but much fewer and less intense compared with the INL (Appendix A). The lower foci number/intensity in the cones compared to the INL might be related to the tissue preparation protocol since 53BP1 signals in cones can be more readily detected in cryosections in comparison to paraffin sections [30].

### 3.6. Rod PRs of Monkeys Show Robust KAP1 Expression

We next compared our findings in rodents and pigs with the situation in diurnal primates. For this we analyzed KAP1 expression levels in the retinae of two macaque species and the common marmoset (*Callithrix jacchus*), which all represent important primate animal models. In all these species we observed robust KAP1 expression levels in the ONL. In Callithrix, cones showed only slightly stronger KAP1 signals compared to rods, whereas in Macaques differences in staining intensities were slightly more pronounced (Appendix A, cones are marked by arrowheads). However, in general a further increase in KAP1 signal strength in rods was observed compared to pigs, which already showed increased signals compared to rodents. IR-induced KAP1 phosphorylation was not analyzed in monkeys since no irradiation experiments of monkeys could be carried out.

### 3.7. Rod PRs of Humans Show Robust KAP1 Expression and Phosphorylation and No Overt DSB Repair Defect

We next analyzed KAP1 expression and phosphorylation levels in PRs of the human retina after in vitro irradiation. The retina of one human eye was isolated and subdivided into several pieces, which were cultivated as retinal explants and irradiated with 1 Gy. As the result of a delay of three hours due to transportation between isolation and irradiation of the samples (see Section 2), we took various control samples. One control sample was fixed directly after retina isolation without cultivation as an explant (Figure 4A,B, referred to as in vivo ctrl). Additional unirradiated controls were fixed at 15 min (in vitro ctrl 15 min) or 8 h (in vitro ctrl 8 h) after isolation (Figure 4C). The analysis showed similar KAP1 staining intensities between rods and cones, an observation unique for the human retina (Figure 4A,D). Staining against pKAP1 showed only a weak signal in the ONL of the in vivo control sample (Figure 4B), while the 15 min in vitro ctrl revealed a surprisingly strong signal. The 8 h in vitro ctrl sample was again weak for pKAP1 (Figure 4C and Appendix A). Of note, KAP1 expression levels were similar for all three ctrl samples as well as for the irradiated samples (Appendix A). To test whether the pKAP1 signal in the 15 min in vitro ctrl sample originated from DSB formation due to transportation, we stained against γH2AX but observed similarly low DSB numbers in all three ctrl samples (Figure 4B,C). In contrast, DSB numbers increased substantially when explants were irradiated with 1 Gy and fixed 15 min later (Figure 4C). IR also increased the pKAP1 signal compared to the 15 min in vitro ctrl sample (Figure 4C and Appendix A). Importantly, no major difference in the pKAP1 staining intensities between rods and cones was observed suggesting that ATM signaling occurs robustly in human rods (Figure 4D). An efficient DSB response was further supported by the robust formation of IR-induced 53BP1 foci, which were not observed in mouse or pig rods (Appendix A). At 8 h after IR, the explants showed a decrease in pKAP1 levels compared to the sample analyzed at 15 min after IR but remained elevated compared to the 8 h in vitro ctrl sample (Figure 4C and Appendix A).

The quantification of γH2AX foci revealed a decrease from 13 foci at 15 min to 7 foci at 8 h in rods and from 12 to 6 foci in cones (Figure 4E). Thus, the DSB repair capacity of PRs in human retinal explants is similar to the situation in pigs (~50% repair within 8 h after IR) but does not show any difference between rods and cones. Taken together, the similar levels of KAP1 expression and phosphorylation as well as the similar capacity to repair IR-induced DSBs between human rods and cones suggests that human rods differ greatly in their response to radiation from rods of nocturnal rodents and diurnal pigs.

## 4. Discussion

GE represents a promising strategy to ameliorate retinal dystrophies [6,7]. During this approach, DSBs are induced into the target cells e.g., by the CRISPR-Cas9 technology and are being processed by the endogenous DSB repair systems. Thus, it is important to understand how repair systems operate in retinal cells. We previously described that rod PRs of adult mice fail to repair ~50% of IR-induced DSBs within 24 h, representing one of the most substantial DSB repair defects observed in WT, i.e., non-mutated cells [8]. Rod PRs of adult mice show a specific inverted chromatin organization and low expression levels of important genes [18,22,23], including ATM and its target protein KAP1 which both have been implicated in the repair of heterochromatic DSBs [8,9,11,12,13]. Since human rod PRs show a conventional chromatin organization, the relevance of this finding for GE strategies remained unclear. Moreover, it remained unclear which model organisms most closely resemble human rods in their response to DSBs. Here, we addressed these questions by analyzing the levels of KAP1 expression, its ATM-dependent phosphorylation and the efficiency of DSB repair in rods of various species with either an inverted or a conventional chromatin organization. Our findings reveal that human rods are distinct to mouse rods and show no sign of impaired KAP phosphorylation and no overt defect in repairing IR-induced DSBs. However, all other species analyzed on our study, including pigs which are frequently employed as a model system, show some degree of KAP1 phosphorylation deficiency and impaired DSB repair. This needs to be considered when projecting results obtained in other species to the situation in humans.

### 4.1. KAP1 Expression and Phosphorylation Levels Are Independent of Chromatin Organization

The rod nuclear inversion in nocturnal animals causes an altered epigenetic landscape [22] and changes the expression of some common proteins [23]. We, therefore, investigated if the failure of mouse rods to express and phosphorylate KAP1 represents a direct consequence of their inverted nuclear architecture. We analyzed Lbr-TER mice which maintain a conventional nuclear organization in rods due to the ectopic overexpression of LBR. Despite the absence of the nuclear inversion, Lbr-TER rods show impaired KAP1 expression and phosphorylation, similar to WT rods. Moreover, pig rods possessing a conventional chromatin organization also show reduced levels of KAP1 expression compared to pig cones. Finally, the diurnal rodent species Arvicanthis, with a chromatin inversion in both rods and cones, shows efficient KAP1 expression and phosphorylation only in cones but not in rods. Thus, KAP1 expression and its IR-induced phosphorylation are impaired in rods without a nuclear inversion and are normal in cones with a nuclear inversion. The uncoupling of KAP1 expression and phosphorylation from chromatin arrangements might be understood from an evolutionary point of view. When some mammalian groups became diurnal and their visual system re-adapted to diurnality, rod PRs in these groups restored the conventional nuclear architecture by restoration of either lam A/C or LBR expression [17,20]. One can speculate that the suppression of KAP1 expression and phosphorylation, which likely was beneficial for nocturnal animals [8,18,21], was recovered only partially after re-adaptation to diurnality and re-acquisition of the conventional nuclear architecture. This can be seen by the increasing amount of KAP1 expression in rods from rodents over pigs to primates and finally humans, the latter being the only ones which show robust KAP1 expression and phosphorylation. This idea is consistent with primates being the earliest mammals in evolution to exhibit strictly diurnal activity [31].

### 4.2. DSB Repair Efficiency in Rod PRs Requires Robust KAP1 Expression and Phosphorylation

Disentangling the level of KAP1 expression and phosphorylation from rod nuclear inversion allowed us to ask if the inefficient repair of DSBs in mouse rods represents a consequence of the failure to express and phosphorylate KAP1 or the nuclear inversion itself. Our results suggest that nuclear inversion has little to no impact while the level of KAP1 expression and phosphorylation determines the DSB repair efficiency. This is substantiated by the observations that (i) genetically manipulated Lbr-TER rods with a conventional chromatin organization fail to express and phosphorylate KAP1 and exhibit a pronounced DSB repair defect; (ii) pig rods with a conventional chromatin organization show reduced KAP1 expression and a reduced DSB repair efficiency compared with pig cones; and (iii) cones of Arvicanthis, which show nuclear inversion similar to their rods, exhibit robust KAP1 expression and phosphorylation and efficiently repair DSBs.

The finding that the presence of KAP1 is required for efficient DSB repair is remarkable since KAP1 has been suggested to represent a barrier for the repair of DSBs in heterochromatin [11,12,13]. However, other studies have suggested an active role of KAP1 in repair after its recruitment to DSBs [14,15]. This is consistent with our observation that KAP1 is recruited to laser-induced DSBs in undifferentiated rods of postnatal mice, in which DSB repair is still efficient [8]. Moreover, KAP1 in the retina is located primarily in euchromatin regions, which is consistent with the idea that it exerts an active function during DSB repair instead of representing a barrier to the process [14,15]. Thus, further studies on the role of euchromatic KAP1 and its phosphorylation (e.g., by the use of phosphomimic KAP1 mutant mice) are necessary to gain a better understanding of its role in DSB repair.

### 4.3. Human Rods Are Distinct from Rods of Other Species

Since efficient DSB repair in rods requires robust KAP1 expression and phosphorylation, our finding that human rods show the highest level of KAP1 expression of all species analyzed is of great importance. Indeed, we observed similar levels of KAP1 expression in human rods and cones while all other species, including pigs and monkeys, displayed reduced levels of KAP1 in their rods compared to their cones. Furthermore, human rods repaired DSBs similarly to human cones and showed 53BP1 foci formation which was not observed in pig rods. Thus, human rods deviate significantly in their response to DSBs compared to mice and pigs although the latter species represent current model systems for the development of GE strategies to prevent rod degeneration causing inherited blinding diseases in humans. We suggest that the described differences between retinae of humans and other species have to be taken into consideration when evaluating the efficiencies of GE strategies. Moreover, the finding that mice, pigs, and human rods repair DSBs differently raises the question of whether other DNA damages (e.g., oxidized bases or cyclobutane pyrimidine dimers) are also differently repaired. If this would be the case, it might possibly explain the observed differences in the susceptibility of rod photoreceptors from various species against light stress or other DNA damaging agents [26]. Thus, further research on DNA damage repair in rod PRs might contribute to the identification of better animal models not only for GE but also for the understanding of the impact of environmental stress (e.g., light stress) on this cell type.

## Figures and Tables

**Figure 1 cells-09-00947-f001:**
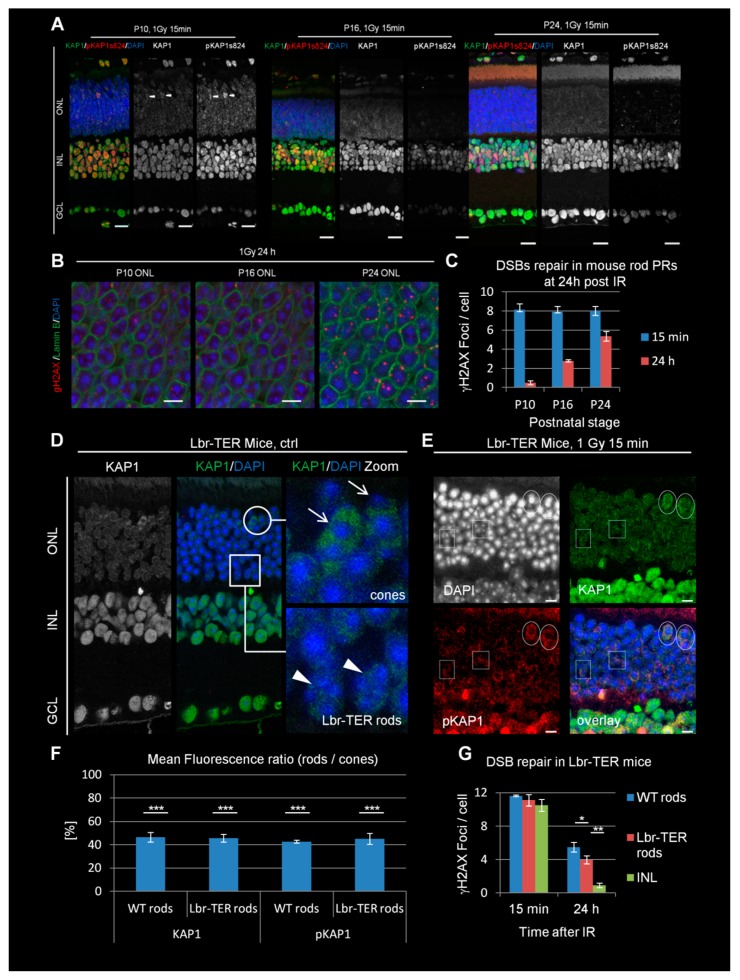
Double-stranded break (DSB) signaling and repair in developing WT and adult Lbr-TER mouse retinae. (**A**) Immunofluorescence images KAP1 (green) and radiation-induced pKAP1 (red) at 15 min after 1 Gy in the retinae of P10, P16, and P24 WT mice. Nuclei were counterstained with 4′-6-Diamidin-phenylindol (DAPI) (blue). Scale bars represent 15 μm. (**B**) Immunofluorescence images of γH2AX (red) at 24 h after 1 Gy in the ONL of P10, P16, and P24 WT mice. Nuclei were counterstained with DAPI (blue) and nuclear borders with lamin B (green). Scale bars represent 5 μm. (**C**) Quantification of residual IR-induced γH2AX foci in rod photoreceptors (PRs) P10, P16, and P24 WT mice at 15 min and 24 h after 1 Gy. (**D**) Immunofluorescence images of KAP1 (green) in the retinae of adult Lbr-TER mice. Scale bar represents 15 µm. Strong KAP1 expressing cones are encircled and marked by arrows in the enlarged pictures. Weakly KAP1 expressing Lbr-TER rods are framed by boxes and marked by arrowheads in the enlarged pictures. (**E**) Immunofluorescence images of KAP1 (green) and radiation-induced pKAP1 (red) in the retinae of adult Lbr-TER mice at 15 min after 1 Gy. Scale bar represents 5 µm. Strong KAP1 expressing cones with IR-induced pKAP1 signals are encircled. Boxes show Lbr-TER rods with weak KAP1 expression and no IR-induced pKAP1 signals. (F) Quantification of the KAP1 and pKAP1 signals in the nuclei of WT and Lbr-TER rods in comparison to cones. The pKAP1 ratio was measured at 15 min after 1 Gy. (**G**) Quantification of residual IR-induced γH2AX foci in WT and Lbr-TER rod PRs at 15 min and 24 h after 1 Gy. All quantitative data are presented as mean ± SEM from three experiments (*n* = 3). *p*-values were obtained by *t*-test and represent a comparison of all cells analyzed in the indicated cell population (at least 40 cells per data point and experiment for foci quantification and 10 nuclei for KAP1/pKAP1 fluorescence measurements in rods and cones) with *: *p* < 0.05; **: *p* < 0.01, and ***: *p* < 0.001. ONL = outer nuclear layer, INL = inner nuclear layer, GCL = ganglion cell layer.

**Figure 2 cells-09-00947-f002:**
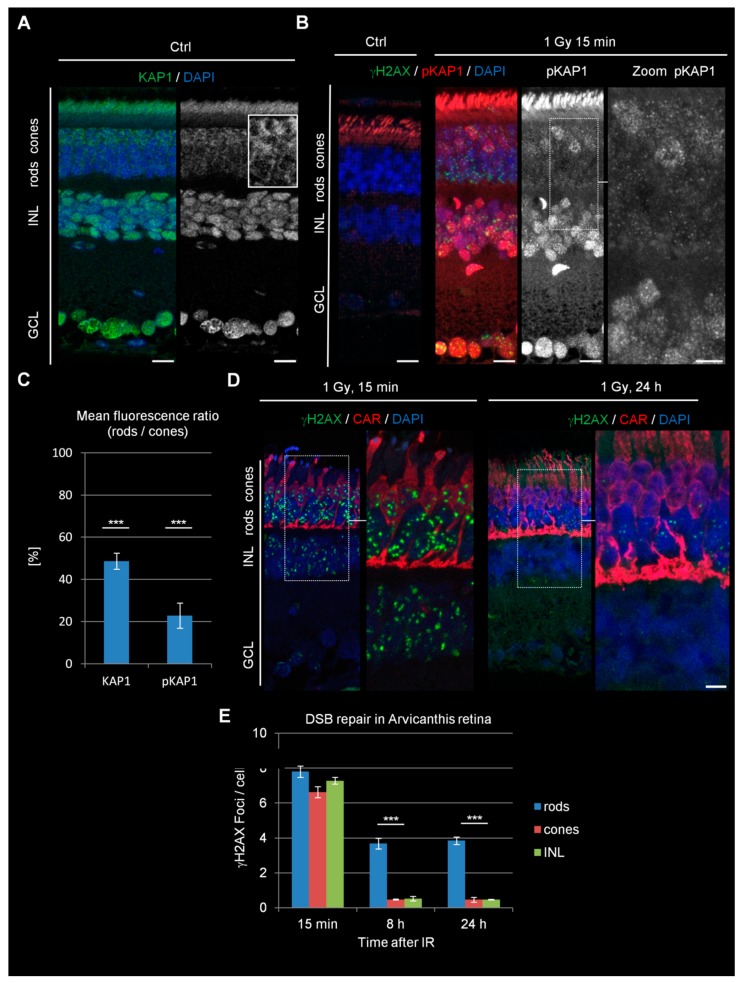
KAP1 expression, DSB signaling, and repair in retinae of adult Arvicanthis ansorgei. (**A**) Immunofluorescence images of KAP1 (green) in the retinae of adult Arvicanthis. Nuclei were counterstained with DAPI (blue). The box shows a higher magnification of rod and cone PRs. Scale bars represent 10 μm. (**B**) Immunofluorescence images of IR-induced γH2AX foci (green) and pKAP1 (red) in ctrl retinae and retinae at 15 min after 1 Gy. Nuclei were counterstained with DAPI (blue). Scale bars represent 10 μm. (**C**) Quantification of the KAP1 and pKAP1 signals in the nuclei of rods compared to cones. The pKAP1 ratio was measured at 15 min after 1 Gy. (**D**) Immunofluorescence images of radiation-induced γH2AX foci (green) and cone arrestin (red) stained cone PRs at 15 min and 24 h after 1 Gy. Nuclei were counterstained with DAPI (blue). Scale bars represent 10 μm. (**E**) Quantification of residual IR-induced γH2AX foci in rods, cones, and INL cells at 15 min, 8 h, and 24 h after 1 Gy irradiation. Data are presented as mean ± SEM from three experiments (*n* = 3). *p*-values were obtained by *t* test and represent a comparison of all cells analyzed in the indicated cell population (at least 40 cells per data point and experiment for foci quantification and 10 nuclei for KAP1/pKAP1 fluorescence measurements in rods and cones) with ***: *p* < 0.001. ONL = outer nuclear layer, INL = inner nuclear layer, GCL = ganglion cell layer, CAR = cone arrestin.

**Figure 3 cells-09-00947-f003:**
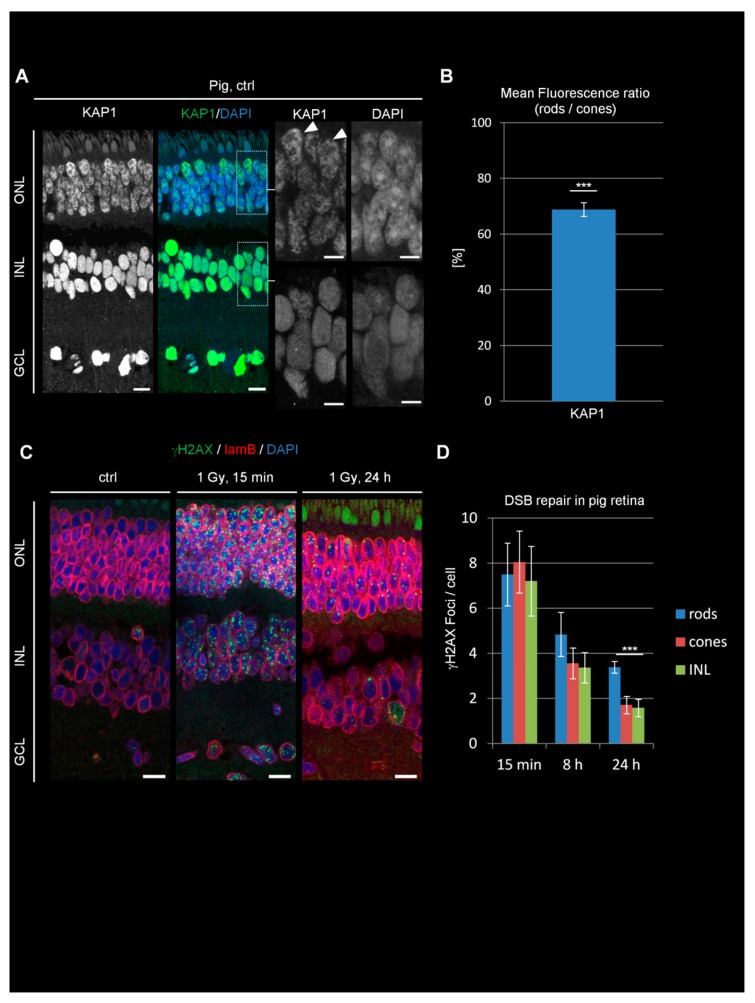
KAP1 expression and DSB repair in pig retinae. (**A**) Immunofluorescence images of KAP1 (green) in pig retinae. Nuclei were counterstained with DAPI (blue). Arrowheads in the enlarged pictures show cone PRs. Scale bars represent 10 μm in whole retina and 5 µm in enlarged pictures. (**B**) Quantification of the KAP1 signal in the nuclei of rods compared to cones. (**C**) Immunofluorescence images of radiation-induced γH2AX foci (green) and lamin B (red) in ctrl and irradiated pig retinae at 15 min and 24 h after 1 Gy. Nuclei were counterstained with DAPI (blue). Scale bars represent 15 μm. (**D**) Quantification of residual IR-induced γH2AX foci in rods, cones, and INL cells at 15 min, 8 h, and 24 h after 1 Gy irradiation. Data are presented as mean ± SEM from three experiments (*n* = 3). Control values are subtracted. *p*-values were obtained by *t* test and represent a comparison of all cells analyzed in the indicated cell population (at least 20 cells per data point and experiment for foci quantification and 10 nuclei for KAP1 fluorescence measurements in rods and cones) with ***: *p* < 0.001. ONL = outer nuclear layer, INL = inner nuclear layer, GCL = ganglion cell layer, lam B = lamin B.

**Figure 4 cells-09-00947-f004:**
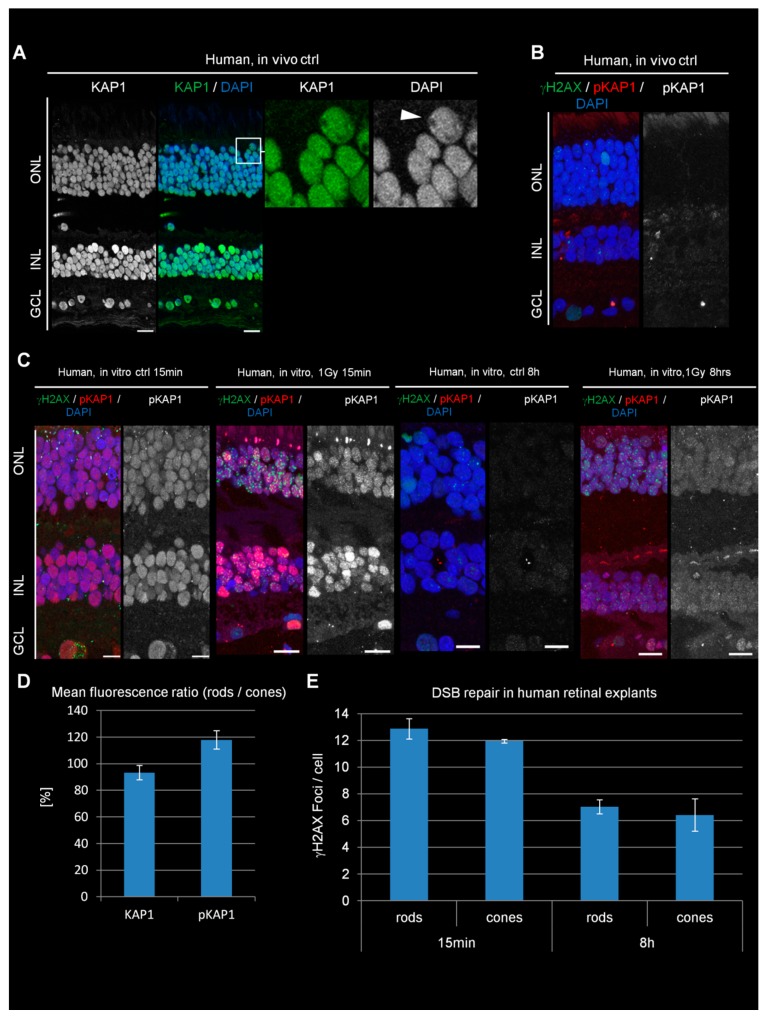
KAP1 expression and its radiation-induced phosphorylation in human retinal explants. (**A**) Immunofluorescence images of KAP1 (green) in human retinal explants. Nuclei were counterstained with DAPI (blue). Scale bars represent 10 μm. (**B**) Immunofluorescence images of spontaneous γH2AX foci (green) and pKAP1 (red) in directly fixed retina (in vivo ctrl). (**C**) Immunofluorescence images of γH2AX foci (green) and pKAP1 (red) in in vitro ctrl and irradiated human retinal explants at 15 min and 8 h after 1 Gy. Nuclei were counterstained with DAPI (blue). Scale bars represent 10 μm. (**D**) Quantification of the KAP1 and pKAP1 signals in the nuclei of rods compared to cones. The pKAP1 ratio was measured at 15 min after 1 Gy. (**E**) Quantification of IR-induced γH2AX foci in rods and cones of retinal explants at 15 min and 8 h after 1 Gy. Data are presented as mean ± SEM from the analysis of three retinal explants for each time point (*n* = 3). Control values are subtracted. *p*-values were obtained by *t* test and represent a comparison of all cells analyzed in the indicated cell population (at least 10 cells per data point and experiment for foci quantification and 10 nuclei for KAP1/pKAP1 fluorescence measurements in rods and cones). ONL = outer nuclear layer, INL = inner nuclear layer, GCL = ganglion cell layer.

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
