# Peer review of "Differences in the Response to DNA Double-Strand Breaks between Rod Photoreceptors of Rodents, Pigs, and Humans"

_cells, 2020, doi:10.3390/cells9040947_

Round 1

Reviewer 1 Report

In this manuscript Frohns et al examine the capacity of rods form different mammalian species to repair gamma-induced DNA double-strand breaks (DSBs). Overall, the data are convincing and the paper is well written. These new results are very interesting and important. I support its publication but have a few comments for the authors to consider before publication. 

  1. Homologous recombination is not restricted to the G2 phase since it is essential to protect and/or restart stressed or collapsed replication forks (line 55-56).
  2. Taking into account several previous reports it should be clarified what does the Authors mean when conclude that: “We and others have previously shown that fully differentiated rod PRs of adult mice lack expression of the of the major stress kinase ataxia-telangiectasia mutated (ATM) (…)” (Line 64-65)? (no transcript? no translation? alternatively spliced form? kinase-dead form?).
  3. There are no immunofluorescence images of untreated control for Fig. 1A.
  4. Are anti-pKAP1 antibodies as specific and efficient in recognizing phospho-Ser824 KAP1 in pig cells as they are in human cell?
  5. It would be important to investigate if similar repair deficiency occurs in mouse rods in the presence of other types of DNA damage e.g. cyclobutane pyrimidine dimers (CPD) or oxidized base pairs. This types of DNA damage occur in heterochromatin and are much more frequent in the genome under normal conditions than DSBs . In other words: is repair inefficiency in mouse rods limited only to DSBs?
  6. It would be interesting to speculate, how rods manage to survive for long periods of time in the presence of multiple DSBs? Would not the accumulation and persistence of broken DNA over time impair more and more transcription?
  7. This is likely beyond the scope of the current paper but Authors could consider construction of KAP1 version with Ser824 to Asp824 mutation that mimics phosphorylation and examine how such rods respond to gamma irradiation (maybe employment of ON/OFF promoter could be helpful).

Author Response

Response to Reviewer 1

First and foremost, we would like to thank the editor and the reviewers for their appreciation of our work and for their useful comments which have substantially improved our paper, thanks.

Below, we provide a point-by-point response to each of the comments raised by the reviewer. We have highlighted the areas in the manuscript where major revisions have been made.

Comments of Reviewer 1:

Reviewer 1: Homologous recombination is not restricted to the G2 phase since it is essential to protect and/or restart stressed or collapsed replication forks (line 55-56).

Answer: The wording was changed into: “Unlike NHEJ, HR uses a sister chromatid as a template and thus repairs DSBs in post-replicative regions of DNA in S and G2 phase [4].”

Reviewer 1: Taking into account several previous reports it should be clarified what does the Authors mean when conclude that: “We and others have previously shown that fully differentiated rod PRs of adult mice lack expression of the of the major stress kinase ataxia-telangiectasia mutated (ATM) (…)” (Line 64-65)? (no transcript? no translation? alternatively spliced form? kinase-dead form?).

Answer: In order to give the reader more information about the status of ATM expression and activity in rod cells we changed the text as followed: “We and others have previously shown that fully differentiated rod PRs of adult mice repair DSBs very inefficiently [8,9]. Of note, these cells fail to efficiently express the DNA damage response kinase ataxia-telangiectasia mutated (ATM), although an alternative splicing form seems to exist [9]. Also the DSB-induced ATM autophosphorylation is strongly impaired in adult mouse PRs, suggesting a strong diminution of ATM kinase activity [8].”

Reviewer 1: There are no immunofluorescence images of untreated control for Fig. 1A.

Answer: Pictures for KAP1 and pKAP1 staining of untreated P10, P16 and P24 mice are now included in Fig. S1A.

Reviewer 1: Are anti-pKAP1 antibodies as specific and efficient in recognizing phospho-Ser824 KAP1 in pig cells as they are in human cell?

Answer: We tested several pKAP1 antibodies that worked well in both human and mice. None of these antibodies was able to detect any signal in the irradiated pig samples, neither in rods nor in any other cells of the retina (which show strong signals in all other species). Thus, we conclude that these antibodies are not specific for the recognition of pKAP1 in pig cells and hence limited our analysis of pig rods to the KAP1 signal.

Reviewer 1: It would be important to investigate if similar repair deficiency occurs in mouse rods in the presence of other types of DNA damage e.g. cyclobutane pyrimidine dimers (CPD) or oxidized base pairs. This types of DNA damage occur in heterochromatin and are much more frequent in the genome under normal conditions than DSBs. In other words: is repair inefficiency in mouse rods limited only to DSBs?

Answer: We absolutely agree that the investigation of the repair capacity of DNA damages other than DSBs in rods is of great importance since it would contribute to a better understanding of the impact of environmental stress (e.g. light stress) on this cell type. However, an analysis of such repair processes would not only require additional animal experiments but also the establishment of the required experimental procedures. Since our lab is specialized in the measurement of DSB-induced signaling and repair events and none of the authors of this manuscript that were responsible for the experimental part are still present in the lab, we are not able to address this point within a reasonable time period.

However, in order to stress the importance of such analysis we included the following paragraph in the discussion: “Moreover, the finding that mice, pigs and human rods repair DSBs differently raises the question of whether other DNA damages (e.g. oxidized bases or cyclobutane pyrimidine dimers) are also differently repaired. If this would be the case, it might possibly explain the observed differences in the susceptibility of rod photoreceptors from various species against light stress or other DNA damaging agents [26]. Thus, further research on DNA damage repair in rod PRs might contribute to the identification of better animal models not only for GE but also for the understanding of the impact of environmental stress (e.g. light stress) on this cell type.”

Reviewer 1: It would be interesting to speculate, how rods manage to survive for long periods of time in the presence of multiple DSBs? Would not the accumulation and persistence of broken DNA over time impair more and more transcription?

Answer: We agree that this is an interesting question. We addressed it already in our previous work where we discussed the fact that adult PRs become very resistant against DSB-induced apoptosis during their postnatal differentiation period. Since ATM is responsible for DSB-induced apoptosis it is not surprising to see that the sensitivity against DSB-induced apoptosis decreases as the expression levels of ATM and its signaling diminishes. However, there is still the question remaining why, despite the DSB repair defect, the level of spontaneous foci is relatively low. Our previous studies showed that breaks which are unrepaired after 24 h can still be repaired at later times. Thus, the repair capacity of rod PRs might still be sufficient to remove breaks with slow kinetics before transcription is impaired and thus cell death is induced. Since we already discussed this in our previous publication we would like to refrain from repeating this issue in this manuscript.

Reviewer 1: This is likely beyond the scope of the current paper but Authors could consider construction of KAP1 version with Ser824 to Asp824 mutation that mimics phosphorylation and examine how such rods respond to gamma irradiation (maybe employment of ON/OFF promoter could be helpful).

Answer: We agree that these experiments would contribute to a better understanding of the role of KAP1 and its phosphorylation in DSB repair. To underline this idea, we included the following sentence in the manuscript: “Thus, further studies on the role of euchromatic KAP1 and its phosphorylation (e.g. by the use of phosphomimic KAP1 mutant mice) are necessary to gain a better understanding of its role in DSB repair.”

Reviewer 2 Report

In this study, Frohns et al. compared the DSB repair efficiency in rod receptors among various species. Overall, this is an interesting attempt and the information they present will be useful for subsequent studies to develop a strategy for intervening the retinal diseases especially by genome editing techniques. However, some of the data they showed were lacking quantification or adequate controls and hardly support their conclusion.

Major points:

Page 6 lines 224-226: We observed that the numbers of unrepaired breaks at 24h after IR increased from 1 in P10 to 3 in P16 and 5 in P24 mice (Fig. 1C).

The authors should show the results at early time points to make sure that the number of DSBs induced is at a similar level.

Page 6 lines 238-240: However both Lbr-TER and WT rods exhibited similar level of KAP1 expression, strongly reduced in comparison to cone PRs (Fig. 1D).

The authors should show the quantitative data to support this argument, e.g. mean intensity of KAP1 staining per nucleus.

Page 6 line 242: the defect in KAP1 expression and phosphorylation arises independently of the Lbr downregulation and nuclear inversion in rods.

The statements presented here by the authors are confusing. Although they argue that “...phosphorylation of KAP1 at ser824 as a typical read-out for active ATM signaling” above, according to their data shown here, the reduction of p-KAP1 seems due to the reduction of KAP1 expression levels. Is ATM signaling in rods defective? If so, how can the author explain the data that γH2AX, which is also heavily dependent on ATM signaling after IR, is proficient in rods?

Page 7 lines 283-284: we assessed the formation of 53BP1 foci at DSBs as an alternative process that is directly dependent on ATM signaling [24,25].

Again, it is still unclear whether the authors stick on the model that ATM signaling is defective in rodent rods or rather that the KAP1 expression is somehow reduced. 53BP1 focus formation is affected by many factors e.g. ubiquitin signaling. Even if ATM signaling is proficient, we can expect many situations that 53BP1 doesn’t form foci.

Page 7 lines 285-286: At 15 min after IR, all cells of the INL showed robust 53BP1 focus formation while rods were almost completely devoid of 53BP1 foci.

The authors should show the 53BP1 foci analysis for mouse rods in order to use these data in pigs to conclude that the rods in pig are in a similar situation to mouse rods.

Page 8 lines 321-322: The 8h in vitro ctrl sample was again negative for pKAP1 (Fig. 4C).

Did the authors test KAP1 levels in the same samples? Did it behave in a similar way as pKAP1? Also, the quantification of these results must be shown.

Minor points:

Page 2 lines 52-54: The CRISPR-Cas9 technique relies on the endonuclease Cas9 to generate DSBs at a specific site of the genome with the subsequent repair of the induced breaks by the endogenous DSB repair machinery [2].

The authors should acknowledge the recent progress and efforts in the genome editing field to use non-endonuclease versions of Cas9 and/or its orthologs.

Page 2 lines 55-56: Unlike NHEJ, HR ensures high fidelity repair by the use of a sister chromatid as a template and thus is restricted to the G2 phase [3,4].

HR can happen in post-replicative regions during S phase.

Page 2 line 62: In order to improve HR efficiency in postmitotic cells, detailed...

The logic here is not clear. If cells are postmitotic, and non-replicating, HR doesn’t occur anyway.

Page 5 line 164: DAPI

The authors need to define abbreviation.

Page 17 line 475:

The authors need to add appropriate information for Supplementary Materials.

Throughout the manuscript, one space between numbers and units should be added.

Author Response

Response to Reviewer 2

First and foremost, we would like to thank the editor and the reviewers for their appreciation of our work and for their useful comments which have substantially improved our paper, thanks.

Below, we provide a point-by-point response to each of the comments raised by the reviewer. We have highlighted the areas in the manuscript where major revisions have been made.

Comments of Reviewer 2:         

Major points:

Reviewer 2: Page 6 lines 224-226: “We observed that the numbers of unrepaired breaks at 24 h after IR increased from 1 in P10 to 3 in P16 and 5 in P24 mice (Fig. 1C).” The authors should show the results at early time points to make sure that the number of DSBs induced is at a similar level.

Answer: A quantification of DSB induction at 15 min after 1 Gy was included in the analysis and is now shown in Fig. 1C. No differences in DSB induction between the three different developmental stages were observed.

Reviewer 2: Page 6 lines 238-240: “However, both Lbr-TER and WT rods exhibited similar level of KAP1 expression, strongly reduced in comparison to cone PRs (Fig. 1D).” The authors should show the quantitative data to support this argument, e.g. mean intensity of KAP1 staining per nucleus.

Answer: Quantification of both KAP1 and pKAP1 signals in Lbr-TER and WT rods are now included in Fig. 1F. The results confirm that Lbr-TER and WT rods exhibit similar levels of KAP1 expression and phosphorylation.

Reviewer 2: Page 6 line 242: “the defect in KAP1 expression and phosphorylation arises independently of the Lbr downregulation and nuclear inversion in rods.” The statements presented here by the authors are confusing. Although they argue that “...phosphorylation of KAP1 at ser824 as a typical read-out for active ATM signaling” above, according to their data shown here, the reduction of p-KAP1 seems due to the reduction of KAP1 expression levels. Is ATM signaling in rods defective? If so, how can the author explain the data that γH2AX, which is also heavily dependent on ATM signaling after IR, is proficient in rods?

Answer: We agree with the reviewer that our statements regarding the role of ATM signaling on KAP1 phosphorylation was confusing. We have now clarified this in the introduction and have re-written all the sections that include this topic. In short, our previous as well as the present study showed that adult mouse PRs exhibit reduced KAP and p-KAP levels. However, the reduction in pKAP is usually greater than the reduction in KAP and we attribute this difference to the reduction in ATM kinase activity. Thus, both the reduced ATM activity as well as the reduced KAP1 protein level contributes to the repair defect in adult rods. This conclusion is also supported by our new quantification of KAP and pKAP levels as suggested by this reviewer.

Although ATM signaling is defective, we don’t expect this to impact on H2AX phosphorylation since DNA-PK and ATM can redundantly phosphorylate H2AX in non-cycling cells (Stiff et al, ATM and DNA-PK function redundantly to phosphorylate H2AX after exposure to ionizing radiation. Cancer Res. 2004 Apr 1; 64(7):2390-6.). In fact, we have shown in our previous paper that rods from ATM knock-out mice show the same number of gH2AX foci as wild-type mice and the same number of gH2AX foci as in other cell types

Reviewer 2: Again, it is still unclear whether the authors stick on the model that ATM signaling is defective in rodent rods or rather that the KAP1 expression is somehow reduced. 53BP1 focus formation is affected by many factors e.g. ubiquitin signaling. Even if ATM signaling is proficient, we can expect many situations that 53BP1 doesn’t form foci.

Answer: We agree that situations other than reduced ATM signaling can impair 53BP1 foci formation and have removed all statements that the lack of 53BP1 foci formation is directly related to a lack of ATM signaling.

Reviewer 2: Page 7 lines 285-286: “At 15 min after IR, all cells of the INL showed robust 53BP1 focus formation while rods were almost completely devoid of 53BP1 foci.” The authors should show the 53BP1 foci analysis for mouse rods in order to use these data in pigs to conclude that the rods in pig are in a similar situation to mouse rods.

Answer: We have now included images and a quantification of the 53BP1 foci numbers in WT and Lbr-TER rods in Fig. S2.

Reviewer 2: Page 8 lines 321-322: “The 8 h in vitro ctrl sample was again negative for pKAP1 (Fig. 4C).” Did the authors test KAP1 levels in the same samples? Did it behave in a similar way as pKAP1? Also, the quantification of these results must be shown.

Answer: KAP1 and pKAP1 levels are now presented as images and quantified in all retinal samples in Fig. S5. The results completely confirm the statements given in the manuscript.

Minor points:

Reviewer 2: Page 2 lines 52-54: “The CRISPR-Cas9 technique relies on the endonuclease Cas9 to generate DSBs at a specific site of the genome with the subsequent repair of the induced breaks by the endogenous DSB repair machinery [2].” The authors should acknowledge the recent progress and efforts in the genome editing field to use non-endonuclease versions of Cas9 and/or its orthologs.

Answer: The following statement was added to the manuscript: “Additionally, Cas9 lacking endonuclease activity can be used to selectively perturb genome expression by blocking transcription.”

Reviewer 2: Page 2 lines 55-56: “Unlike NHEJ, HR ensures high fidelity repair by the use of a sister chromatid as a template and thus is restricted to the G2 phase [3,4].” HR can happen in post-replicative regions during S phase.

Answer: The wording was changed into: “Unlike NHEJ, HR uses a sister chromatid as a template and thus repairs DSBs in post-replicative regions of DNA in S and G2 phase [4].”

Reviewer 2: Page 2 line 62: “In order to improve HR efficiency in postmitotic cells, detailed... .” The logic here is not clear. If cells are postmitotic, and non-replicating, HR doesn’t occur anyway.

Answer: We agree that this phrasing was misleading. Originally, we wanted to make the point that researchers are actually trying to establish HR-based genome editing strategies in postmitotic cells. In fact, we actually included a citation of a paper showing that this approach indeed is possible in rod photoreceptors. In order to make this clear we changed the text of the manuscript as followed: “However, in order to make use of the higher accuracy of HR over NHEJ, there are efforts to establish high fidelity GE by HR even in postmitotic cells. In fact, this can be achieved by the combined delivery of a donor template together with the CRISPR-associated endonuclease Cas9 [5]. Using this approach, Bakondi et al. could successfully modify mutated alleles in postmitotic rods of newborn pups of a rat model for rod degeneration-based blinding diseases [6]. However, up to now this approach is only successful in rods of young (postnatal) animals, hindering its translation to human patients, as retinal dystrophies are only diagnosed at late stages of human development. However, various research groups are working on ways to further improve the HR efficiency in postmitotic cells, e.g. by using molecules that are inhibiting NHEJ [7].”

Reviewer 2: Page 5 line 164: DAPI. The authors need to define abbreviation.

Answer: We now provide abbreviations, sorry for this.

Reviewer 2: Page 17 line 475: The authors need to add appropriate information for Supplementary Materials.

Answer: We now provide the correct information.

Reviewer 2: Throughout the manuscript, one space between numbers and units should be added.

Answer: This has been corrected.

Round 2

Reviewer 2 Report

All my concerns were satisfactorily addressed by the authors in this revised manuscript.